# Imbalanced class distribution and performance evaluation metrics: A systematic review of prediction accuracy for determining model performance in healthcare systems

Michael Owusu-Adjei⬤*, James Ben Hayfron-Acquah, Twum Frimpong, Gaddafi Abdul-Salaam

Department of Computer Science, Kwame Nkrumah University of Science and Technology, Kumasi, Ghana

* mowusuadjei@st.knust.edu.gh

**Data Availability Statement:** Since this is a reviewed research article, all relevant secondary source materials used for analysis can be found in

## Abstract

Focus on predictive algorithm and its performance evaluation is extensively covered in most research studies to determine best or appropriate predictive model with Optimum prediction solution indicated by prediction accuracy score, precision, recall, f1score etc. Prediction accuracy score from performance evaluation has been used extensively as the main determining metric for performance recommendation. It is one of the most widely used metric for identifying optimal prediction solution irrespective of dataset class distribution context or nature of dataset and output class distribution between the minority and majority variables. The key research question however is the impact of class inequality on prediction accuracy score in such datasets with output class distribution imbalance as compared to balanced accuracy score in the determination of model performance in healthcare and other real-world application systems. Answering this question requires an appraisal of current state of knowledge in both prediction accuracy score and balanced accuracy score use in real-world applications where there is unequal class distribution. Review of related works that highlight the use of imbalanced class distribution datasets with evaluation metrics will assist in contextualizing this systematic review.

## Author summary

The incidence of unequal class distribution in real-world applications such as healthcare and other non-medical settings continue to receive attention due to machine learning technique challenges with minority class contribution in datasets with imbalanced class distribution. Challenges such as discounting minority class contribution which may be the subject of interest. Predictive modeling evaluation of such datasets with prediction accuracy score which does not take into account dataset class distribution variation could create an erroneous impression of a supposedly high performing machine learning technique as it discounts minority class contribution. Estimating predictive model performance with balanced accuracy score that incorporates other important metrics such as

the relevant journals and web pages detailed in the reference list.

**Funding:** The author(s) received no specific funding for this work.

**Competing interests:** The authors have declared that no competing interests exist.

true positives, true positive rates, true negatives, true negative rates, false positive, false positive rates, false negatives and false negative rates could help assess machine learning model performance more adequately and accurately to determine appropriate model performance.

## 1.0 Introduction

Key component in disease treatment is estimating outcome after treatment is initiated. An outcome is driven mainly by two critical issues; patient response and efficient treatment strategies on the part of healthcare givers. Developing effective and efficient strategies [1] for managing severely ill patients remains a major challenge for healthcare providers. Increasing morbidity and mortality as undesirable consequence of insufficient care practices of uncontrolled blood pressure by individuals. This is an important justification for adopting predictive learning technique capable of identifying important correlated factors associated with the incidence of hypertension. Predictive learning techniques assist in providing real-time solution to low detection rates among many segments of society. Increasing data generation capacity together with available tools necessary for data collection has contributed to the adoption of predictive modeling use in health care systems. Automated systems such as Internet of things (IoT) as an emerging paradigm [2,3] involving human interactions and interconnection of devices has contributed to the availability of large volumes of datasets being witnessed today. Characteristically, healthcare systems are associated with generation of large volumes of datasets brought on by connected medical device use such as remote patient monitoring and virtual assistant device for blood pressure, pulse, heart rate, diabetic monitors etc. Other connected devices include, connected contact lenses, glucose monitors, wearable, fitness tracking devices, virtual healthcare assistants, virtual dispensing assistants etc. Data generated from these applications have been explored in many research works to identify patterns of change using different predictive machine learning (ML) approach including non-clinical [4] to enhance disease diagnosis for improved treatment outcome. Assessing predictive modeling performance has become focused in many research works that includes review studies on feature selection methods and predictive model use in lung cancer radio mics [5]. This study found random forest and support vector machine useful in classification tasks in review studies investigated. Additionally, the use of environmental parameters to improve deep learning model performance for the prediction of COVID-19 daily cases in 9 cities across three countries in different climatic zones using a variety of recurrent neural networks (LSTM) concludes that the inclusion of environmental parameters resulted in improved model performance [6]. Diabetes prediction with applied data mining techniques such as random forest, support vector machines, logistic regression and naïve bayes showed that logistic regression achieved the highest prediction accuracy score of 82.46% as compared to others [7]. Comparative study on model performance in predictive modeling of cardiac arrest in smokers using heart rate variability parameter proved that applying random forest technique achieved the best prediction accuracy score of 93.61% against 88.50% for logistic regression and 92.59% for decision tree classifier [8].

Evaluation in general involves three important qualities which are systematic, assessment and the determination of value, worth and significance. Systematic connotes an interpretation which is structured to give meaning. Different predictive techniques include the use of different or same evaluation metrics [9]. Example, predictive evaluation metrics for ML techniques in classification analysis may be the same or differ from those used in regression analysis depending on the problem under consideration. The challenge here is when to use what and

for what reason and to what benefit. Identifying the appropriate domain for use and for what reason such as evaluate performance for optimization or estimating the number of correctly classified patients for treatment default, number of patients with certain types of diseases etc could provide better use of predictive models. In this review, we offer a thorough discussion on various performance evaluation metrics in line with key research question: Effects of using prediction accuracy score as compared to balanced accuracy to determine appropriate machine learning model for predictive performance in datasets with unequal class distributions (imbalanced datasets) predominant in real-world applications.

## 1.1 Related works

It is important that the development and evaluation of ML techniques are made transparent and interpretable to allay any doubt about its usability in healthcare systems. Predictive model evaluation especially in healthcare and other real-world application systems with class distribution inequality must take into account the peculiarity of the dataset especially when assessing predictive model performance [10]. Prediction accuracy score show results obtained from both observed and predicted values. It is predominantly used in classification problems where there are no dataset class imbalance and no skewed class examples. However one of the challenges identified in many research works is its use as the main performance metric to estimate best or appropriate machine learning model technique in real world applications such as healthcare systems where dataset class distribution inequality is prevalent. The challenge of using prediction accuracy as a measure of model performance is mentioned in a related review work that examined the prospects of machine learning use in clinical outcomes [11]. Concerns regarding prediction accuracy score use is shared in a study of disease diagnosis with 20 machine learning techniques comprising Naïve Bayes, Support Vector Machine (SVM), K-Nearest Neighbors (KNN), Perceptron, Light Gradient Boosting Machine, extreme Gradient Boosting which addressed this challenge with f1-score evaluation metric [12]. Prediction accuracy score obtained ranged between 49%-77%with various techniques but f1-score obtained ranged between 47%-82%.s

Review study of artificial intelligence in disease diagnosis mentioned prediction accuracy as one of the evaluation parameters of interest [13]. Similarly, comparative study of disease prediction with supervised ML techniques also identified prediction accuracy score as performance metric [14]. Similar use of prediction accuracy [15] in assessing best ML technique for breast cancer prediction recorded an accuracy score of 98.7% for techniques such as decision trees and other ensemble techniques. ML principles and applications in real world systems have also been explored [16]. Automatic prediction system for diabetic patients with several ML techniques for explainable artificial intelligence [17] concluded with prediction accuracy score of 81% and auc score of 84%. Additional studies to predict pressure ulcer nursing adverse event [18] using four ML techniques; decision trees, Support Vector Machines, Random Forest and Artificial Neural Networks achieved prediction accuracy score of 94.94% for Support vector machine, 97.93% for Decision trees, 99.88% for Random Forests and 79.02% for Artificial Neural Networks. Determination of appropriate ML algorithms to identify mental health problems [19] in its early stage with techniques such as Logistic Regression, Gradient Boosting, Neural Networks, K-Nearest Neighbor, Support Vector Machine and ensemble techniques showed overall prediction accuracy score of 88.80% achieved by Gradient Boosting. Additional studies to predict heart disease with ML algorithms such as K-Nearest Neighbors (KNN), Naive Bayes and Random Forest singled out Random Forest as the best performing classifier with prediction accuracy score of 95.63% [20]. Further studies for ML use in cardiovascular disease prediction with learning techniques such as support vector machine, convolutional

neural networks and boosting classifiers produced prediction roc_auc score of range 81%-97% [21]. Diagnosis of breast cancer with learning techniques such as linear discriminant analysis (LDA) and Support vector machine (SVM) for various roles had prediction accuracy reading of 99.2% and 79.5% [22]. However, the prediction of breast cancer with Decision tree and Random forest techniques [23] showed prediction accuracy score of 91.18% and 95.72% respectively. Additional ML application as decision support [24] for the detection of breast cancer through feature selection with ML techniques K-Nearest Neighbor, linear discriminant analysis and probabilistic neural network yielded accuracy score of 99.17%. Furthermore [25], prediction of breast cancer with ML based framework using ML techniques; Random Forest, Gradient Boosting, Support Vector Machine, Artificial Neural Network, and Multilayer Perception to achieve better classification accuracy using correlation-based feature selection together with recursive feature elimination extraction resulted in prediction accuracy score of 99.12%. Similarly, with weighting feature and backward elimination feature selection approach [26], application of Random forest ML technique to create computer-aided diagnostic system to distinguish breast cancer tumor between malignant and benign yielded prediction accuracy score of 99.7% and 99.82% respectively. Achieving higher precision and prediction accuracy using K-fold cross-validation with all features in model 2, all features without validation in model 1, with feature selection for model 3 and feature selection together with cross-validation [27] for model 4 using ML techniques; logistic regression, support vector machines, Naive Bayes, Decision trees and k-nearest neighbor, produced different prediction accuracy score at each stage. Highest accuracy score of importance recorded were; 98.83% for support vector machine, 97.17% for K-Nearest Neighbor and 97.88% for Logistic regression. Similarly, ML based model for early stage heart disease prediction with techniques support vector machine, K-nearest neighbor, random forest, Naive Bayes and decision tree using feature selection techniques (chi-square, ANOVA, and mutual information) to determine best fit model concluded that Random forest had the highest prediction accuracy score of 94.51% [28].

Related study for choice of best ML model for prediction of [29] breast cancer also had prediction accuracy score of 98% for Artificial Neural Network, 98% for Decision tree classifier, 99% for K-Nearest Neighbor, 98% for Logistic regression and 100% for Support vector machine. Risk prediction and diagnosis [30] of breast cancer through a comparative analysis of ML techniques to assess model efficiency and effectiveness with respect to prediction accuracy, precision, sensitivity and specificity proved that support vector machine had the highest prediction accuracy performance of 97.13% with the least error rate. Related study [31] to predict and diagnose breast cancer using ML techniques and to determine best model with evaluation metrics such as confusion matrix, accuracy and precision proved that Support Vector Machine among other ML techniques (Random Forest, Logistic Regression, Decision tree (C4.5) and K-Nearest Neighbors) achieved the greatest prediction accuracy score of 97.2%. The continuous use of models such as Support vector machines, Logistic regression and Random forest and Clustering in classification problems such as chronic disease diagnosis is emphasized in a related study that found them to be useful [32]. Similarly, the prediction of treatment trend for patients suffering from hypothyroidism using sodium levothyroxine with ML techniques showed that using extra-trees achieves better prediction accuracy of 84%. [33]. Following from this [34] is a predictive study of chronic kidney disease prediction with three ML techniques namely; Random forest, Support Vector machine and Decision tree together with recursive feature elimination technique. This study showed different prediction accuracy score in situations where feature selection is used and others where feature selection is not used. Prediction accuracy recorded with feature selection techniques were as follows; 99.8% for Random forest, 95.5% for Support vector machine and 98.6% for Decision tree. Additional studies on predictive modeling of chronic diseases such as sclerosis progression over 6 and 10

year period using ML techniques [35] such as K-nearest neighbor, Support vector machine, Decision tree and Logistic regression concluded with performance evaluation metric area under the curve score (auc), sensitivity, specificity, geometric mean and f1-score for each period and auc score for disease severity in the 6[th] year are KNN 74%, Decision tree 74%, Linear regression 80% and Support vector machine 80%. Disease severity in the 10[th] year had auc score KNN 67%, Decision tree 57%, Linear regression 67% and Support vector machine 73%.

Furthermore studies [36] for the detection of chronic kidney disease to show important correlations or predictive attributes using ML techniques (k-nearest neighbors, random forest, and neural networks) and 24 features used accuracy, root mean squared error (rmse) and fi-score measure as evaluation parameters. Predicted accuracy score of 99.3%forRandom forest classifier was achieved. Additional research to identify advanced chronic kidney disease with ML techniques; generalized linear model network, random forest, artificial neural network and natural language processing [37] showed improved prediction performance in accuracy score as reported. Prediction accuracy score for ML techniques used were; both for training data and testing data: Logistic regression 81.8% and 81.9%, Random forest 91.3% and 82.1%, Decision tree 86.0% and 82.1%. Its conclusion recommends improvement on achieved prediction accuracy score. Application of deep learning technique for prediction and classification of hypertension with related variables [38] showed the following prediction accuracy scores; Deep neural network: (75%, 73.9%, 74.3%, 74.3%) and Decision tree: (67.6%, 68.4%, 69%, 68%). Related study [39] on the prediction of hypertension using features such as patient demographics, past and current patient health condition and medical records for the determination of risk factors using artificial neural network showed prediction accuracy score of 82%.

Understanding disease symptoms is one sure way of effectively controlling and managing its treatment outcome. Predictive modeling [40] of heart disease risks and its symptoms using ML techniques will ensure effective patient care. Implementation of heart disease risk prediction using six ML techniques (support vector machine, Gaussian Naive Bayes, Logistic regression, light gradient boosting model, extreme gradient boosting and Random forest) showed the following predicted accuracy score; 80.23%, 78.68%, 80.32%, 77.04%, 73.77% and 88.5% respectively.

A population level-based approach [41] for predicting hypertension using ML techniques (extreme Gradient Boosting, Gradient Boosting Machine, Logistic Regression, Random forest, Decision tree and Linear Discriminant Analysis) had predicted accuracy score of 90% for (extreme Gradient Boosting, Gradient Boosting Machine, Logistic Regression and Linear Discriminant Analysis) as compared to 89% for Random forest and 83% for Decision tree.

**1.1.0 Accuracy score in non-health settings.** Related research perspectives in other real-world applications such as spam message detection, fraud detection and risk estimation/forecasting are explored in this section. The risk of spam messaging and its impact on business operations are far reaching some of which include hacked systems and ransom demand payments, destruction of critical data and infrastructure and many others. Applying effective, efficient ML modeling technique that identifies important characteristics for the detection and subsequent prevention or destruction of threats posed continue to engage research attention. A study to detect spam threats [42] in emails and IoT platforms using Naïve Bayes, decision trees, neural networks and random forest together with other techniques had prediction accuracy score and precision score as follows; for Suppost Vector Machine and Naive Bayes 96.9%, precision 93.12% and Naive Bayes; 99.46%, precision 99.66%. Similarly, transformer-based embedding with ensemble learning techniques for spam detection showed prediction accuracy score of 99.91% [43]. Furthermore application [44] of hybrid algorithm for the detection of malicious spam messaging in email with ML techniques Naive Bayes, Support vector machines, Logistic Regression and Random Forest showed predicted accuracy score of 96.15%

for Naive Bayes, 96.15% for support vector machine, 98.08% for Logistic regression and 95.38% for Random forest respectively. Evaluation of automatic short message service performance [45] using Naive Bayes, BayesNet, C4.5, J48, Self-organizing map and Decision tree showed predicted accuracy score of 89.64%, 91.11%, 80.24%, 79.2%, 88.24% and 75.76% respectively. Comparative performance evaluation to improve prediction accuracy [46] of two ML models; support vector machine and random forest for the detection of junk mail spam showed prediction accuracy of models as; Support vector machine 93.52% and Random forest 91.41%.Related to improving prediction accuracy is the issue of improving training time and reducing prediction error rate. ML based hybrid bagging technique application [47] using random forest and decision tree (J48) for the analysis of email spam detection showed 98% prediction accuracy score. Other performance metrics evaluated include true negative rates, false positive rate and false negative rate, precision, recall and f-measure (f1-score). Increase in online transactions including online payments has also increased the risk of credit card fraud, ML based credit card fraud detection system [48] using genetic algorithm with the following learning techniques (Decision Tree, Random Forest, Logistic Regression, Artificial Neural Network, and Naive Bayes showed that applied genetic algorithm feature selection led to a predictive accuracy score of 100% for both Decision tree and Artificial neural network. Related to study [48] is financial fraud detection system in healthcare using ML techniques such as deep learning to address the challenge of credit card fraud monitoring [49]. Applying ML techniques (Naive Bayes, Logistic Regression, K-Nearest Neighbor, Random Forest, and Sequential Convolutional Neural Network) resulted in the predicted accuracy score; 96.1%, 94.8%, 95.89%, 97.58%, and 92.3% respectively. Strategies have been adapted and adopted to deal with the challenge of fraud detection by various organizations. One such solution is provided by [50] which implemented ML based self-analyzing system to flag potential fraudulent activities for review. Case study approach [51] for a review of ML techniques (logistic regression, decision tree, random forest, K-Nearest Neighbor and extreme Gradient Boosting) in credit card fraud detection evaluated best model prediction performance using accuracy, recall, precision and f1score metrics. The study identified Logistic regression and K-nearest Neighbor as best performing classifiers. Implementation of fraud detection tools [52] to identify anomalies on financial applications using outlier detection techniques such as Local outlier factor, Isolation factor and Elliptic envelope and ML techniques (Random forest, Adaptive boosting and extreme gradient boosting) showed predicted accuracy score of 99.95%.Modeling [53] of medical visits by patients suffering from diabetes with ML techniques; logistic regression, support vector machine, linear discriminant analysis, quadratic discriminant analysis, extreme gradient boosting, neural networks and deep neural network obtained balanced accuracy score of 65.7%. Similarly, predicting length of stay [54] from admission to clinical ward with ML techniques random forest, decision trees, support vector machine, multi-layer perceptron, adaboost and gradient boost concluded with random forest as the best performing technique with balanced accuracy score of 72% at the initial stage of admission and 75% in-admission. However, an up-sampling approach [55] for breast cancer prediction using k-nearest neighbor, decision tree, random forest, neural networks, support vector machine and extreme gradient boosting obtained balanced accuracy score of 97.47%.

**1.1.1 Related works summary.** Systematic review of related research works had key objectives and among them was the search for literature with the following characteristics; a focus on current state of knowledge with respect to ML techniques, applications and evaluations, research works with prediction accuracy score as an evaluation metric, research works in real-world context with unequal class distributions using relevant methodologies. Excluded from this review article search were defining specific search timeline and the motivation for not specifying search period was to include as many important related works as possible

irrespective of its date of publication. Of particular interest was work on healthcare systems and other real-world applications (spam detections, fraud predictions, risk predictions etc). A summary of identified characteristics among selected reviewed literature with emphasis on prediction accuracy score as performance metric is presented in Table 1.Literature search sources were; Google scholar and other online journal databases such as IEEE, puhmed, hindawi journals, BioMed central, Pmc, Elsevier, Sciencedirect, organizational websites, online libraries and many other journals. A total of 80 articles were screened for (relevancy) and determined inclusion criteria was for related works in healthcare practice that had used predictive machine learning either in disease diagnosis, prediction, risk or treatment assessment. Literature of related works with ML applications in other relevant settings such as spam detection in mails, sms spamming were also considered. No time frame exclusion criteria was used, but about 80% of selected materials were mainly published works between 2016–2022 and a handful in 2023.Observations noticed in related literature used indicate extensive use of ML techniques in real-world applications for various reasons including serving as decision support systems. Predominantly used techniques include Random forest, Support vector machine, Logistic regression, K-Nearest Neighbor, Decision trees, Gradient boosting classifier and few ensemble techniques. The use of evaluation performance metrics such as precision, recall, f1-score, prediction accuracy and in some instance predicted positive and predicted negative values is observed. Of interest is the use of prediction accuracy as a predominant metric for assessing model performance found among all the related literature reviewed.

**1.1.2 Strengths and weaknesses identified in reviewed literature.** In many of the literature reviewed, the pattern of high prediction accuracy score is observed including the use of more than one predictive technique modeling for comparative analysis. The use of predictive modeling in disease detection, diagnosis and treatment outcome for diseases of public concern together with predictive modeling in e-mail spam predictions, fraud detections, risk predictions etc is also observed. The desire for many is to address challenges with novel techniques from different perspectives. Differences in feature selection and optimization technique tools use to estimate variable importance and to improve on prediction performance is also indicated with varying outcome. Model performance evaluation is also indicated in almost all literature reviewed. However, there is a strong desire with few exceptions among majority of the reviewed literature to estimate best model performance significantly on prediction accuracy score irrespective of the problem domain and dataset class distribution. We also note the recorded high value of balanced accuracy score by [55] achieved using up-sampling optimization technique from [53,54].

## 1.2 Research question

The incidence of dataset class inequality in most real-world applications including healthcare systems and how it affects predictive modeling performance has received little attention in current research studies. Minority class contribution which is overlooked by most learning algorithms in such situations is rarely addressed by related research works resulting in skewed model performance evaluation estimate influenced mainly by the majority class contribution. As an example; in the prediction of patient treatment default; the number of non-defaulters may far exceed the number of defaulters by 100s of 1000s or in significant ratio such as 1:100000 but the key challenge is to correctly identify minority patient defaulters for necessary interventions. Therefore assessing model performance within this context with prediction accuracy score creates a challenge for better model performance assessment as minority class contribution is discounted.

**Table 1. Reviewed literature descriptions.**

| Reference no | Research type | Methodology | Evaluation metric | Score value (%) |
|---|---|---|---|---|
| [8] | Disease diagnosis | Naïve Bayes, Support Vector Machine (SVM), K-Nearest Neighbors (KNN), perceptron, Light Gradient Boosting Machine, extreme Gradient Boosting. | accuracy | 49–77 |
| [12] | Breast cancer prediction | Decision trees, ensemble techniques | accuracy | 98.7 |
| [14] | Diabetic prediction | | | 81 |
| [15] | Pressure ulcer nursing prediction | Decision trees, Support Vector Machines, Random Forest and Artificial Neural Networks | accuracy | 94.94, 97.93, 99.98, 79.02 |
| [16] | Identify mental health | Logistic Regression, Gradient Boosting, Neural Networks, K-Nearest Neighbor, and Support Vector Machine, as well as an ensemble techniques | accuracy | 88.80 |
| [17] | Heart disease prediction | K-Nearest Neighbors (KNN), Naive Bayes and Random Forest singled out Random Forest | accuracy | 95.63 |
| [19] | Cardiovascular disease prediction | support vector machine, convolutional neural networks boosting classifiers | accuracy | 81–97 |
| [20] | Breast cancer diagnosis | linear discriminant analysis (LDA) and Support vector machine (SVM) | accuracy | 99.2, 79.5 |
| [21] | Breast cancer prediction | Decision tree, Random forest | accuracy | 91.18, 95.72 |
| [22] | Detection of breast cancer | K-Nearest Neighbor, linear discriminant analysis | accuracy | 99.17 |
| [23] | Prediction of breast cancer | Random Forest, Gradient Boosting, Support Vector Machine, Artificial Neural Network, and Multilayer Perception | accuracy | 99.12 |
| [24] | Differentiate between malignant and benign tumors diagnosis | Random forest with weighting and backward elimination feature techniques | accuracy | 99.7, 99.82 |
| [26] | early stage heart disease prediction | Support vector machine, K-nearest neighbor, Random forest, Naive Bayes, and Decision tree | accuracy | 94.51 |
| [27] | Prediction of breast cancer | Artificial Neural Network, Decision tree classifier, K-Nearest Neighbor, 0.98, Logistic regression, Support vector machine | accuracy | 98, 98, 99, 98, 100 |
| [28] | Breast cancer risk prediction | Support vector machine | accuracy | 97.13 |
| [29] | Breast cancer diagnosis | Support vector machine, Random Forest, Logistic Regression, Decision tree (C4.5) and K-Nearest Neighbors | accuracy | 97.2 |
| [31] | Treatment trend prediction for hypothyrodism | Extra trees | accuracy | 84 |
| [32] | Chronic kidney disease prediction | Random forest, support vector machine, decision tree | accuracy | 99.8, 95.5, 98.6 |
| [33] | Chronic disease progression (sclerosis) | K-nearest neighbor, support vector machine, decision tree, logistic regression | auc | |
| [34] | Chronic kidney disease detection | K-nearest neighbor, random forest, neural networks | accuracy | 99.3 |
| [35] | Advanced chronic kidney disease prediction | Logistic regression, random forest, decision tree | accuracy | 81.9, 82.1, 82.1 |
| [36] | Prediction of hypertension | Deep neural network, decision tree | accuracy | 75, 69 |
| [37] | Risk prediction (hypertension) | k-nearest neighbor, multi-layer perceptron | accuracy | 82.47 |
| [38] | Prediction of hypertension | Artificial neural network | accuracy | 82 |
| [39] | Heart disease risk prediction | support vector machine, Gaussian Naive Bayes, Logistic regression, light gradient boosting model, extreme gradient boosting and Random forest | accuracy | 80.23, 78.68, 80.32, 77.04, 73.77, 88.5 |
| [40] | Hypertension prediction | extreme Gradient Boosting, Gradient Boosting Machine, Logistic Regression, Random forest, Decision tree and Linear Discriminant Analysis | accuracy | 83–90 |
| [41] | Spam detection | Naive Bayes, decision tree, neural networks, random forest, support vector machine | accuracy | 96.9–99.66 |
| [42] | Spam detection | ensemble | accuracy | 99.91 |
| [43] | Malicious spam in mails | Naive Bayes, support vector machine, logistic regression and random forest | accuracy | 96.15, 96.15, 98.08, 95.38 |
| [44] | Sms spam classification | Naive Bayes, BayesNet, C4.5, J48, Self-organizing map and Decision tree | accuracy | 89.64, 91.11, 80.24, 79.2, 88.24, 75.76 |
| [45] | Junk email detection | Support vector, random forest | accuracy | 93.52, 91.41 |
| [46] | Email spam detection | Bagging, random forest, decision tree (J48) | accuracy | 98 |
| [44] | Credit card fraud detection | Decision Tree, Random Forest, Logistic Regression, Artificial Neural Network, and Naive Bayes | accuracy | 100 |

*(Continued)*

**Table 1.** (Continued)

| Reference no | Research type | Methodology | Evaluation metric | Score value (%) |
|---|---|---|---|---|
| [45] | Financial fraud detection in healthcare | Naive Bayes, Logistic Regression, K-Nearest Neighbor, Random Forest, and Sequential Convolutional Neural Network | accuracy | 96.1, 94.8, 95.89, 97.58, and 92.3 |
| [48] | identify anomalies on financial applications | Random forest, Adaptive boosting and extreme gradient boosting | accuracy | 99.95 |
| [49] | Unplanned medical visit | Logistic regression, support vector machine, neural network, deep neural network, extreme gradient boosting, linear discriminant analysis, quadratic discriminant analysis | Balance accuracy | 65.7 |
| [50] | Patient length of stay | Random forest, decision tree, support vector machine, multi-layer perceptron, adaboost and gradient boosting | Balanced accuracy | 75 |
| [51] | Breast cancer prediction | k-nearest neighbor, random forest, decision tree, neural network, support vector machine and extreme gradient boosting | Balanced accuracy | 97.47 |

### 1.3 Research objective

This research therefore explores other evaluation metrics that takes into account dataset class inequality to estimate reasonable prediction accuracy score for the determination of best or appropriate predictive technique performance. We therefore propose a novel evaluation approach for predictive modeling evaluation in healthcare systems context called Proposed Model Evaluation Approach (PMEA) which addresses minority class contribution challenges in predictive modeling. It is derived in combination with two most important evaluation metrics (True positive rates and True negative rates: TPR, TNR) to estimate more accurately best or appropriate model performance in context.

### 2.0 Materials and methods

A systematic review of related research works through an adopted search strategy protocol for relevant literature with focus on characteristics such as current state of knowledge with respect to ML techniques, applications and evaluations, research works with prediction accuracy score as an evaluation metric, research works in real-world context with appropriate methodologies. Excluded from this review search were defining specific search timelines and the motivation for not specifying search period was to include as many important works as possible irrespective of publication date. Of particular interest were related works on healthcare systems and other real-world applications (spam detections, fraud predictions, risk predictions etc) with dataset class distribution inequality.

Our approach was to adopt guidelines emphasized in the preferred reporting items for systematic reviews and meta-analyses (PRISMA) protocol. These protocols were; designing the research question, adopting searches and search strategy, developing inclusion and exclusion criteria, designing data extraction plan to synthesis and draw conclusions, quality assessment criteria rule and developing strategies to analyzed the collected data.

**2.0.1 Search strategy.** Literature used was obtained from the following sources; PuMed, Google scholar, Web of science indexed journals, Scopus indexed journals (Springer nature, Hindawi, Elsevier, ScienceDirect, IEEEAccess, IEEEXplore) and many others. Search words included; predictive modeling in healthcare systems, machine learning prediction accuracy score, disease diagnosis with machine learning, machine learning prediction of disease (chronic kidney, hypertension, breast cancer, machine learning model performance evaluations, fraud detection with machine learning, detection of spam messages with machine learning, machine learning prediction with balanced accuracy score, dealing with class imbalance

in machine learning etc. Our search period started from 2016 to ensure access to most materials since ML use in healthcare has been limited since its inception.

**2.0.2 Inclusion criteria.** Our inclusion criteria for relevant articles was; model performance evaluation metrics, evaluation with accuracy scores, prediction with ML methods (techniques), ML applications in healthcare, ML use in healthcare (diagnosis, treatments, disease management), fraud detections, spam detections, risk predictions, junk mail predictions, ML in disease treatment default, deep learning applications in healthcare and many others.

**2.0.3 Exclusion criteria.** Excluded from the search criteria was; ML application articles without performance evaluation, articles considered to be outside the realm of real-world application, articles with duplicate findings, articles with findings inconsistent with stated research objectives and reviewed articles.

**2.0.4 Data extraction plan.** To assist in extracting relevant information from the sourced documents, every single article downloaded were placed in Mendeley Desktop including source documents from non-academic websites including industrial webpages with relevant information.

**2.0.5 Quality assessment.** Our quality assessment procedure was to follow through with all protocols stated in PRISMA guidelines and this resulted in the use of 68.6% of total articles sourced meeting all inclusion and exclusion criteria.

## 2.1 Evaluation metrics in classification

Brief description of performance evaluation metrics used in most machine learning applications for classification to demonstrate metric use and reasons for its use.

**2.1.1 Prediction accuracy.** In ML, prediction accuracy defines how well a model performs at predictions on unseen data. Prediction accuracy is only a fraction of model predictions that are correct [56]. Prediction accuracy is illustrated as

$$\text{Accuracy} = \frac{\text{Number of correct Predictions}}{\text{Total number of Predictions}}$$

Subsequently in classification, accuracy is calculated in terms of positive and negative predictions.

$$\text{Accuracy} = \frac{TP + TN}{TP + TN + FP + FN}$$

Where

TP = True positives, TN = True negatives, FP = False positives, FN = False negatives

**2.1.2 Receiver operating characteristic curve (roc_auc).** roc_auc measures the performance of ML model's ability to differentiate between classes. A higher roc_auc curve score closer to 1 indicates favorable model performance at predicting 0 as 0 and 1 as 1. Some of the terms used in roc_auc curve are TPR (True positive rates/ Recall/Sensitivity)

$$\text{TPR/Recall/Sensitivity} = \frac{TP}{TP + FN}$$

$$\text{Specificity} = \frac{TN}{TN + FP}$$

$$\text{FPR} = \frac{FP}{TN + FP}$$

Where FPR = False positive rates

## ROC score curve

Decrease in threshold leads to increase in more positive values and an increase in sensitivity with a decrease in specificity. Conversely an increase in threshold leads to more negative values and a high specificity with low sensitivity [57].

**2.1.3 Confusion matrix.** Classification performance metric which consists of combination of predicted and actual values is the foundation on which precision, recall, roc_auc, specificity and prediction accuracy is derived.

**2.1.4 Log-loss.** Log-loss measures the closeness of the prediction probability to the corresponding actual value or true value (0 or 1). A higher log-loss is indicative of divergence of the prediction probability from the actual value.

**2.1.5 Precision.** Precision refers to the identification of relevant data points, its ability to identify true data points that are positive and classified by the model also as positive. False negative predictions are data points the model identifies as negative but are truly positive (false alarm).

Precision = TP/(TP + FP)

Where TP = true positives

FP = false positives

**2.1.6 Recall.** A models ability to identify all relevant class instances in a dataset. In certain situations, precision and recall can be combined to achieve optimal solution to a problem such as identifying all patients labeled as defaulters to disease treatment. This will lead to a high recall value but a low precision score.

**2.1.7 F1 score.** F1 score is the harmonic mean of precision and recall that achieves optimal solution (combining precision and recall). It is the weighted mean average of precision and recall and used extensively in search engines for relevant information retrieval.

F1 score = 2* (precision * recall)/(precision + recall)

## 2.2 Evaluation metrics in regression

Some of the evaluation metrics used in regression analysis are as follows;

- Mean Squared Error (MSE)

- Mean Absolute Error (MAE)

- Mean Absolute Percentage Error (MAPE)

- Root Mean Squared Error (RMSE)

- Root Mean Error (RME)

- Adjusted R-Squared (Adjusted $R^2$)

**2.2.1 Balanced accuracy.** A metric used in imbalanced datasets for evaluation performance. It is the average of sensitivity and specificity.

Balanced accuracy = $\underline{TPR + TNR}$

Where

TPR = True Positive Rates

TNR = True Negative Rates

## 3.0 Results

Performance evaluation metrics used in model assessments such as receiver operating characteristic curves, confusion matrix that describes misclassifications, average prediction accuracy score determination (balanced accuracy) is presented in Fig 1, Fig 2 and Fig 3 respectively.

ROC Curve

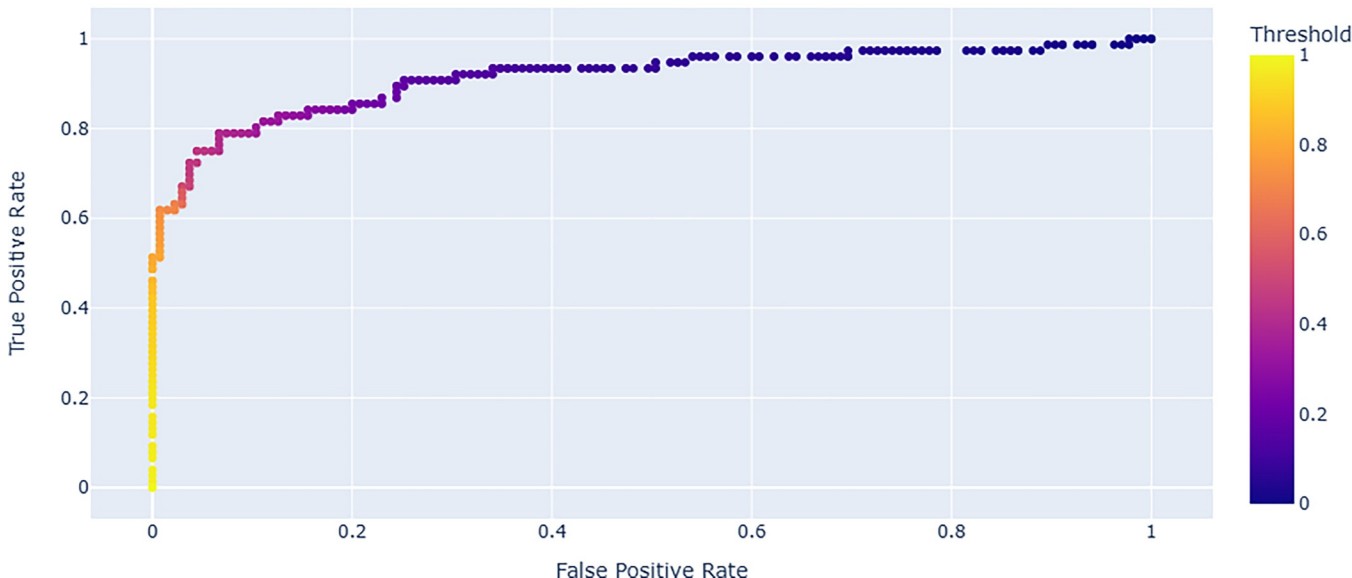

**Fig 1. Receiver operating characteristic curve.** roc_auc score curve. This figure shows predictive modeling performance at different thresholds. Predictive model ability to differentiate between classes is measured using roc_auc score curve.

## Selection of related works process flowchart

PRISMA guidelines for data collection including statistical observations is simplified in Fig 4 which describes literature selection processes, criteria, period of collection, exclusion and

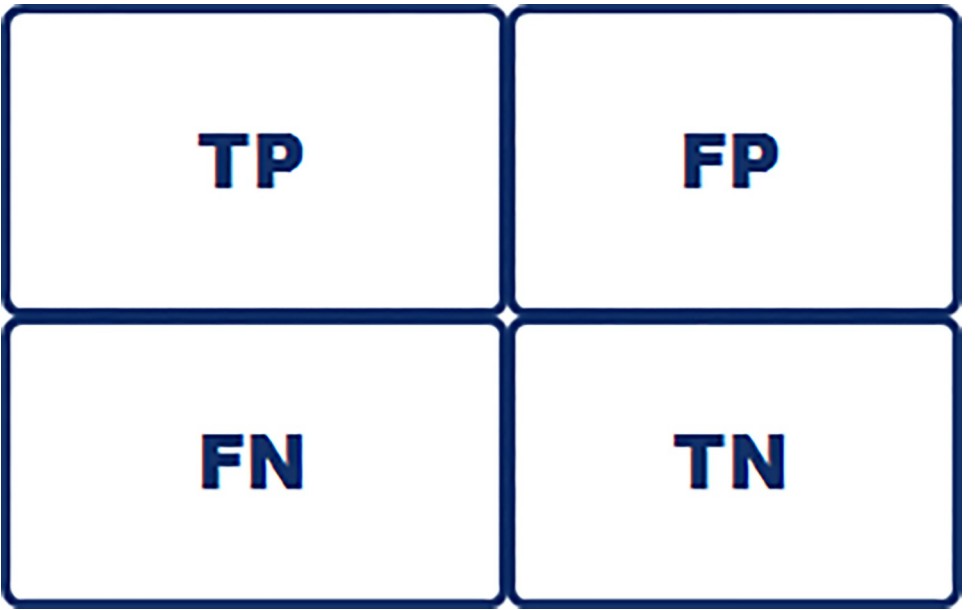

**Fig 2. Classification performance metric.** Confusion matrix. Made up of predicted values and actual values presented to show how many true positive predicted values, true negative predicted values, false positive predicted values and false negative predicted values were obtained by each predictive model.

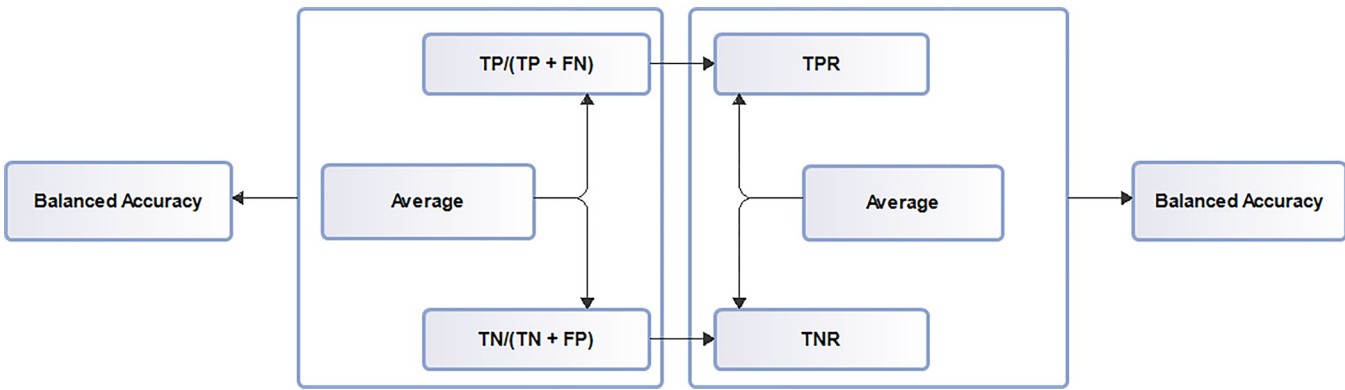

**Fig 3. Accuracy prediction.** Balanced accuracy diagram. Determining balanced accuracy involves the determination of other important performance indices such as true positives, true negative, true positive rates, true negative rates, false positives, false negatives, false positive rates, false negative rates which are necessary to assess model performance regarding class distribution inequality.

inclusion criteria, selected number of articles used with distribution share of each criteria also presented in Fig 5 and Fig 6.

## Balanced accuracy score estimation

Balanced accuracy score determination is presented in a flowchart diagram detailing other evaluation metrics that address minority class contribution such as true positives, true negatives, false positives, false negatives, true negative rates, true positive rates, false positive rates and false negative rates is displayed is presented in Fig 7

## Evaluation model

Flowchart display evaluation model that addresses minority class contribution showing how balanced accuracy score is achieved together with other metrics that constitute false alarm is also presented.

## Discussion

In this study, review of related literature on the use of predictive modeling in real-world applications with dataset class distribution inequality such as healthcare for either prediction of a

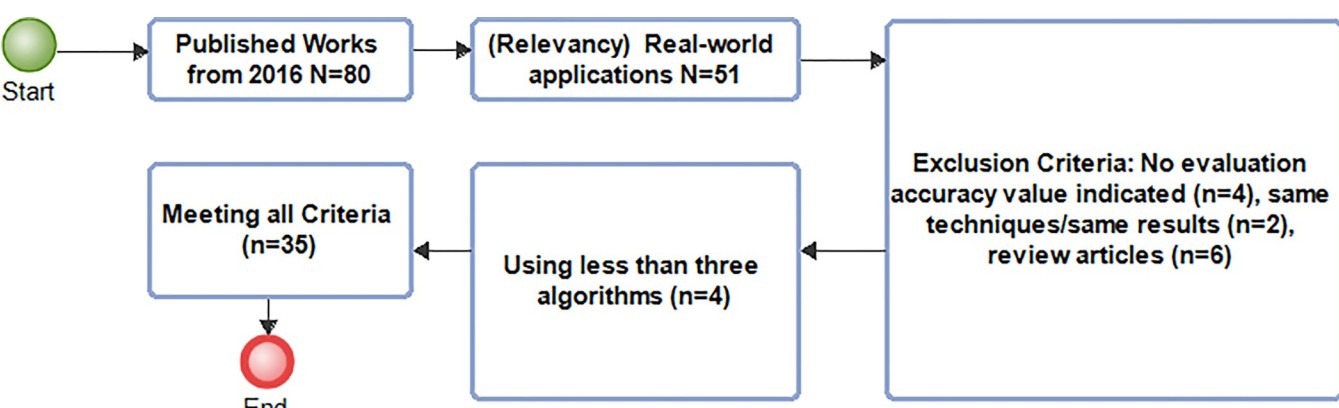

**Fig 4. Methodology.** Related research works selection process flowchart. This is a flowchart diagram of the methodology process showing specific achievable tasks at each stage. Its design is based on guidelines specified in PRISMA 2023.

## Proportions of Relevant and Irrelevant articles in all published works

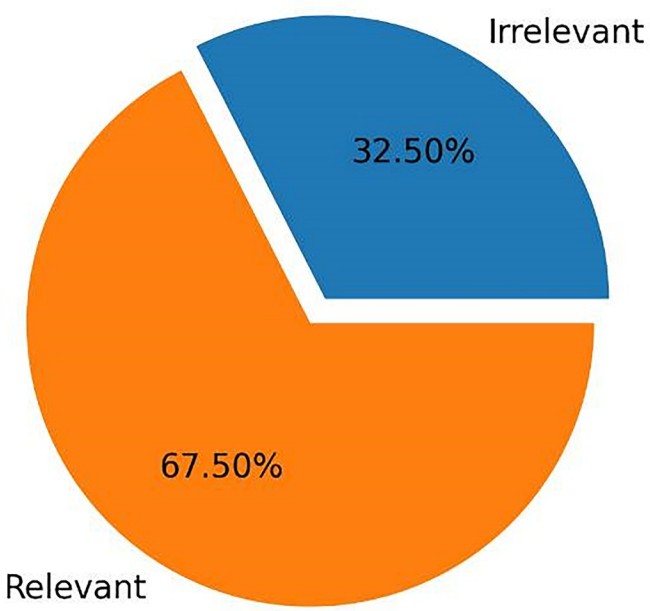

**Fig 5. Relevant and irrelevant distribution share.** Proportion of relevant and irrelevant distribution share. Collected data distribution based on relevancy. Distribution share of relevant and irrelevant related research works is shown in this figure.

certain disease or diagnosis of a disease and its related outcome have been examined. Approaches to estimate prediction outcomes have also been examined in the identified literature. Both strengths and weaknesses identified have been described. Challenges with approaches have been mentioned. This review is not the final determination of all the challenges in ML applications as ML use is diverse and keep expanding. Continuous expansion could bring about new challenges and opportunities. While it maybe fair to use prediction accuracy to justify model performance, its contextual application maybe understood than the generalization of its use as the final evaluation metric to determine best model performance in business applications with dataset class imbalance. Model evaluation to determine performance in healthcare systems playa unique role because lives are at stake. Assessing predictive performance based on probability score of false positives or false negatives (false alarms) within healthcare systems could be more beneficial to estimate best model performance as compared to general use prediction accuracy score. Prediction accuracy estimate based on the number of true positives and true negative rates could be fair justification for estimating best model in performance within healthcare systems as it addresses disparities in output class distributions.

## Contribution

This paper highlights an important ingredient in the choice of best machine learning model for prediction and places this choice under context. We also make an assertion that the supposedly higher prediction accuracy scores as obtained in some research findings with dataset class imbalance when compared with balanced accuracy scores of studies using similar ML techniques in the same context creates an erroneous impression of high performing models among individual ML techniques and for this reason the choice of best performing ML model based on prediction accuracy is problematic if context and purpose for prediction modeling is

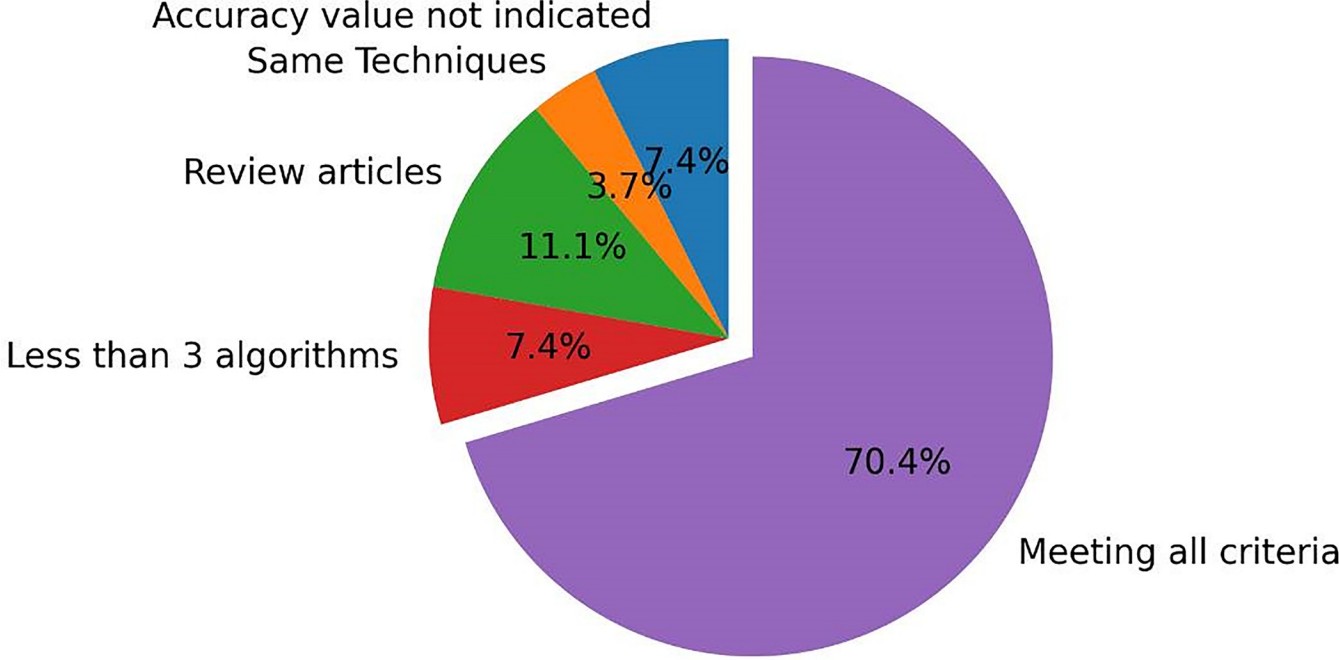

**Fig 6. Distribution of Exclusion criteria.** Proportion of exclusion criteria in relevant articles. Distribution share of exclusion criteria in relevant research works identified is displayed in this figure. Each exclusion criteria distribution share is indicated by the accompanying percentage value.

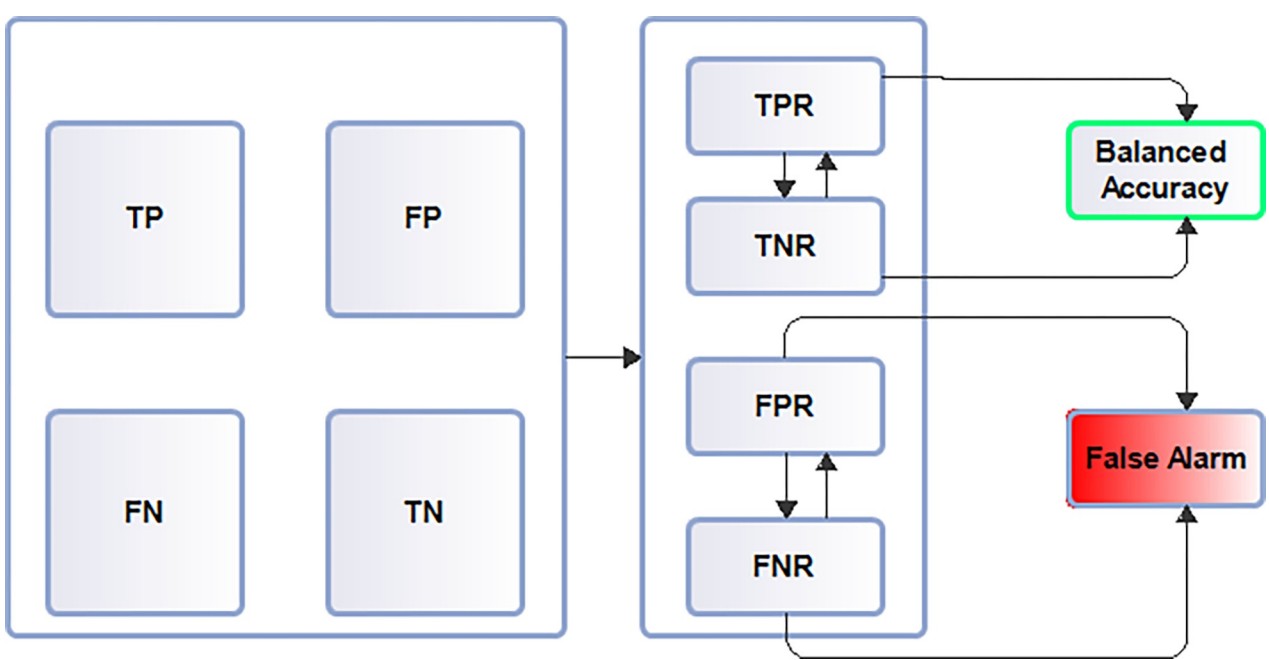

**Fig 7. Evaluation model.** Proposed model evaluation approach. Adopted evaluation approach that addresses dataset class inequality including false alarm rates is presented in this diagram. This approach could help identify best model performance in datasets with class distribution imbalance.

not considered. We have used only one evaluation metric (prediction accuracy score) but many others remain, we therefore encourage further discussions on the appropriate use of all other evaluation metrics for emphasis.

## Conclusion

In the light of challenges identified with the use of prediction accuracy as a performance measure for best model determination with imbalanced dataset, we propose a novel evaluation approach that takes into account dataset class imbalance for predictive modeling use in healthcare systems context called Proposed Model Evaluation Approach (PMEA). PMEA, addresses the use of prediction accuracy as an evaluation performance metric challenge with balanced accuracy score derived from two most important evaluation metrics (True positive rates and True negative rates: TPR, TNR) to estimate model performance in datasets with unequal class distribution which can be generalized in similar contexts. The application of this model to practical business applications could generate more insight into appropriate model choice for enhanced performance Identifying appropriate evaluation metric(s) for performance assessment with imbalanced dataset class distribution will ensure a true determination of best performing prediction model for recommendation in context. We have examined literature, identified individual approaches to solving issues including context and examination of individual approaches. We have proposed an approach to deal with an identified challenge in context. This, we believe is not exhaustive, other evaluation assessments for its applicability in context will be examined in future research studies.

## Supporting information

**S1 PRISMA Checklist. Identification of new studies via databases and registers.** PRISMA checklist indicates processes used in Identifying, screening with inclusion and exclusion criteria of related works evaluated in this study.
(PDF)

## Acknowledgments

We acknowledge the support and cooperation of management and staff of Kwahu Government Hospital.

## Author Contributions

**Conceptualization:** Michael Owusu-Adjei.

**Data curation:** Michael Owusu-Adjei.

**Formal analysis:** Twum Frimpong.

**Investigation:** Michael Owusu-Adjei.

**Methodology:** Twum Frimpong.

**Project administration:** Gaddafi Abdul-Salaam.

**Resources:** Michael Owusu-Adjei.

**Supervision:** James Ben Hayfron-Acquah.

**Writing – original draft:** Michael Owusu-Adjei.

**Writing – review & editing:** Twum Frimpong.

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
