## [Decision Letter · Decision Letter 0]

29 Aug 2023

PDIG-D-23-00204

A systematic review of prediction accuracy as an evaluation measure for determining machine learning model performance in healthcare systems.

PLOS Digital Health

Dear Dr. Owusu-Adjei,

Thank you for submitting your manuscript to PLOS Digital Health. After careful consideration, we feel that it has merit but does not fully meet PLOS Digital Health's publication criteria as it currently stands. Therefore, we invite you to submit a revised version of the manuscript that addresses the points raised during the review process.

Please submit your revised manuscript within 60 days Oct 28 2023 11:59PM. If you will need more time than this to complete your revisions, please reply to this message or contact the journal office at digitalhealth@plos.org. Please include the following items when submitting your revised manuscript:

We look forward to receiving your revised manuscript.

Kind regards,

Anat Reiner-Benaim

Academic Editor

PLOS Digital Health

Journal Requirements:

1. Please provide separate figure files in .tif or .eps format only and remove any figures embedded in your manuscript file. Please also ensure that all files are under our size limit of 10MB.

Additional Editor Comments (if provided):

Reviewers' comments:

Reviewer's Responses to Questions

**Comments to the Author**

1. Does this manuscript meet PLOS Digital Health’s publication criteria? Is the manuscript technically sound, and do the data support the conclusions? The manuscript must describe methodologically and ethically rigorous research with conclusions that are appropriately drawn based on the data presented.

Reviewer #1: Partly

Reviewer #2: No

Reviewer #3: Yes

Reviewer #4: Yes

2. Has the statistical analysis been performed appropriately and rigorously?

Reviewer #1: No

Reviewer #2: No

Reviewer #3: Yes

Reviewer #4: Yes

3. Have the authors made all data underlying the findings in their manuscript fully available (please refer to the Data Availability Statement at the start of the manuscript PDF file)?

Reviewer #1: No

Reviewer #2: No

Reviewer #3: Yes

Reviewer #4: Yes

4. Is the manuscript presented in an intelligible fashion and written in standard English?

Reviewer #1: No

Reviewer #2: No

Reviewer #3: Yes

Reviewer #4: Yes

5. Review Comments to the Author

Reviewer #1: Dear Author,

1. The title of manuscript needs re-synthesis as in its present form it is not acceptable.

2. The methodilogy is vague and too many subtopics in each section of manuscript compromises logical flow of manuscript.

3. Subheading Result part is missing and reults are aglomerated in methodology.

4. The manuscript need to be edited rigorously by native English speaker for English language and grammar

Reviewer #2: There is no universal method that can be considered superior across all diverse datasets. Different methods cannot be directly compared across varying datasets, even when using the same methodology. While formal meta-analyses like the G-test might not exist, it's important to note that comparisons are complex. Lastly, it's worth highlighting that calibrating a model cannot solely rely on AUC or confusion matrices – there are various other approaches to consider.

Reviewer #3: The research findings has been well presented based on the available literature related to the ML algorithms and their advancement in the field of Medical Sciences. With the advancement in AI and ML based medical technologies, proper assessment of the methods is required.

Reviewer #4: Summary

1. The Research article addresses a very modern aspect in the use of digital tools in health. 

2. The methodology is fairly broken down in details with the exact methods stated and elaborated such as the PRISMA protocol which is commonly used for systematic reviews. The inclusion and exclusion criteria of the paper selections are clear enough.

3. The Machine learning knowledge elaborated and expressed in the paper is good and diverse exploring some rarely used metrics such as balanced accuracy.

4. The conclusion is supported by the data and the results presented.

Major issues

I have not identified any major issues in the research article.

Minor issues

1. I would have liked to see research questions or the specific objectives stated or listed together independently in the related works summary as they are not easily noted when spread in the the literature of the related works.

2. Briefly at the categorization of your research as either qualitative or quantitative or even both. From observation, I see some substantial amounts of quantitative methods.

3. The method of specificity was assigned to sensitivity just below the Receiver operating characteristic curve (ROC). Sensitivity = TN / (TN + FP) it should have been specificity equated to that method.

4. It would be better to have the method of precision and fl score elaborated just like the others to help the readers put into perspective how they relate to the confusion matrix.

5. Maintain the same units of percentages % in the section above the “works related Summary” section. I suggest you replace the decimals like 0.98 to percentages for consistence.

6. In the discussion section, in this phrase “(false alarms) within the healthcare system will be more beneficial to estimate best model performance than a focus on prediction accuracy score.” I suggest to replace will be with could be as I feel there is no sufficient evidence in this paper to justify the claim. The false alarms could have different impact once used on different datasets and using different ML models in comparison with the prediction accuracy.

7. In the “Evaluation metrics in Classification” section: “Additional studies on predictive modeling of chronic diseases such as sclerosis progression and outcomes using ML techniques [35] using ML techniques such as K-nearest neighbor”, what were the scores of prediction accuracy of the respective modules in that study. Also typo in repetition of the phrase “ML techniques”.

8. There are some typos and grammar issues in the article which will need to be worked on such as:

a) Some very Long sentences in the introduction section.

b) In the section for “recall”, maintain the same font size, etc

6. PLOS authors have the option to publish the peer review history of their article (what does this mean?). If published, this will include your full peer review and any attached files.

**Do you want your identity to be public for this peer review?** For information about this choice, including consent withdrawal, please see our Privacy Policy.

Reviewer #1: No

Reviewer #2: No

Reviewer #3: No

Reviewer #4: No

---

## [Decision Letter · Decision Letter 1]

25 Oct 2023

PDIG-D-23-00204R1

Imbalanced class distribution and performance evaluation metrics: A systematic review of prediction accuracy for determining model performance in healthcare systems.

PLOS Digital Health

Dear Dr. Owusu-Adjei,

Thank you for submitting your manuscript to PLOS Digital Health. After careful consideration, we feel that it has merit but does not fully meet PLOS Digital Health's publication criteria as it currently stands. Therefore, we invite you to submit a revised version of the manuscript that addresses the points raised during the review process.

Please submit your revised manuscript within 30 days Nov 24 2023 11:59PM. If you will need more time than this to complete your revisions, please reply to this message or contact the journal office at digitalhealth@plos.org. Please include the following items when submitting your revised manuscript:

We look forward to receiving your revised manuscript.

Kind regards,

Anat Reiner-Benaim

Academic Editor

PLOS Digital Health

Journal Requirements:

Additional Editor Comments (if provided):

Reviewers' comments:

Reviewer's Responses to Questions

**Comments to the Author**

1. If the authors have adequately addressed your comments raised in a previous round of review and you feel that this manuscript is now acceptable for publication, you may indicate that here to bypass the “Comments to the Author” section, enter your conflict of interest statement in the “Confidential to Editor” section, and submit your "Accept" recommendation.

Reviewer #4: All comments have been addressed

Reviewer #5: All comments have been addressed

Reviewer #6: (No Response)

2. Does this manuscript meet PLOS Digital Health’s publication criteria? Is the manuscript technically sound, and do the data support the conclusions? The manuscript must describe methodologically and ethically rigorous research with conclusions that are appropriately drawn based on the data presented.

Reviewer #4: Yes

Reviewer #5: Yes

Reviewer #6: Partly

3. Has the statistical analysis been performed appropriately and rigorously?

Reviewer #4: Yes

Reviewer #5: Yes

Reviewer #6: I don't know

4. Have the authors made all data underlying the findings in their manuscript fully available (please refer to the Data Availability Statement at the start of the manuscript PDF file)?

Reviewer #4: Yes

Reviewer #5: Yes

Reviewer #6: No

5. Is the manuscript presented in an intelligible fashion and written in standard English?

Reviewer #4: Yes

Reviewer #5: (No Response)

Reviewer #6: Yes

6. Review Comments to the Author

Reviewer #4: 1. All my concerns have been well addressed.

2. I like the new breakdown of the sections all the way from the abstract, introduction, literature, research questions, research objectives, materials and methods, until the conclusion.

3. Fl score formula has a very small typo “an existing underscore _”. Kindly remove it to make the formula be like F1 score = 2* (precision * recall) / (precision + recall), maybe the editor may also be able to correct it.

Reviewer #5: 1. The topic of research is unique and fits in to contemporary needs of health care industry 

2. In Introduction Section: Replace "patient compliance" with "patient response"

3. Introduction Section: Give examples for usage of predictive modeling performance in current research works with appropriate references.

4. Section 1.1.0 ( Accuracy score in non-health settings) - This section doesn't provide info on whether examples mentioned in the section consist of datasets with unequal class distributions or not?

Reviewer #6: 1. If the authors have adequately addressed your comments raised in a previous round of review and you feel that this manuscript is now acceptable for publication, you may indicate that here to bypass the “Comments to the Author” section, enter your conflict of interest statement in the “Confidential to Editor” section, and submit your "Accept" recommendation. 

I have not made comments in previous rounds, so this field remained as "Please select response".

2. Does this manuscript meet PLOS Digital Health’s publication criteria? Is the manuscript technically sound, and do the data support the conclusions? The manuscript must describe methodologically and ethically rigorous research with conclusions that are appropriately drawn based on the data presented.

This manuscript is original, high importance and of broad interest, and methodological and ethical. However, I selected "partly" because the related work was interesting, but it would have been more impactful to briefly discuss how this new model/framework can help the minority population in the conclusion or discussion.

3. Has the statistical analysis been performed appropriately and rigorously? 

"I don't know" was not selected due to minimal expertise.

4. Have the authors made all data underlying the findings in their manuscript fully available (please refer to the Data Availability Statement at the start of the manuscript PDF file)?

"No" was selected due to this section not being filled out in the manuscript.

5. Is the manuscript presented in an intelligible fashion and written in standard English?

"Yes" was selected because the manuscript was interesting and easy to read.

7. PLOS authors have the option to publish the peer review history of their article (what does this mean?). If published, this will include your full peer review and any attached files.

**Do you want your identity to be public for this peer review?** For information about this choice, including consent withdrawal, please see our Privacy Policy. 

Reviewer #4: Yes: Hillary Kaluuma

Reviewer #5: No

Reviewer #6: No

---

## [Editor Report · Decision Letter 2]

29 Oct 2023

Imbalanced class distribution and performance evaluation metrics: A systematic review of prediction accuracy for determining model performance in healthcare systems.

PDIG-D-23-00204R2

Dear Mr. Owusu-Adjei,

We are pleased to inform you that your manuscript 'Imbalanced class distribution and performance evaluation metrics: A systematic review of prediction accuracy for determining model performance in healthcare systems.' has been provisionally accepted for publication in PLOS Digital Health.

Best regards,

Anat Reiner-Benaim

Academic Editor

PLOS Digital Health